# Mo_2_C-Loaded Porous Carbon Nanosheets as a Multifunctional Separator Coating for High-Performance Lithium–Sulfur Batteries

**DOI:** 10.3390/ma16041635

**Published:** 2023-02-15

**Authors:** Jianli Zhang, Yang Wang, Zhenkai Zhou, Qiang Chen, Yiping Tang

**Affiliations:** College of Material Science and Engineering, Zhejiang University of Technology, Hangzhou 310014, China

**Keywords:** lithium–sulfur batteries, shuttle effect, Mo_2_C/C, polar catalyst, separator modification

## Abstract

Lithium–sulfur batteries have emerged as one of the promising next-generation energy storage devices. However, the dissolution and shuttling of polysulfides in the electrolyte leads to a rapid decrease in capacity, severe self-discharge, and poor high-temperature performance. Here, we demonstrate the design and preparation of a Mo_2_C nanoparticle-embedded carbon nanosheet matrix material (Mo_2_C/C) and its application in lithium–sulfur battery separator modification. As a polar catalyst, Mo_2_C/C can effectively adsorb and promote the reversible conversion of lithium polysulfides, suppress the shuttle effect, and improve the electrochemical performance of the battery. The lithium–sulfur battery with the Mo_2_C/C =-modified separator showed a good rate of performance with high specific capacities of 1470 and 799 mAh g^−1^ at 0.1 and 2 C, respectively. In addition, the long-cycle performance of only 0.09% decay per cycle for 400 cycles and the stable cycling under high sulfur loading indicate that the Mo_2_C/C-modified separator holds great promise for the development of high-energy-density lithium–sulfur batteries.

## 1. Introduction

The energy storage technology revolution is in full swing. As the cost of traditional lithium-ion batteries increases and their energy density approaches the theoretical limit, economical energy storage devices with high energy density have attracted much attention [1,2,3,4,5]. Among the available battery systems, lithium–sulfur (Li-S) batteries based on multi-electron conversion chemistry have satisfactory theoretical energy density (2600 Wh kg^−1^) and gravimetric specific capacity (1675 mAh g^−1^) and are considered to be a promising candidate [6,7,8,9,10]. Moreover, the abundant reserves and environmental friendliness of cathode sulfur materials ensure the commercial potential of Li-S batteries [11,12,13]. However, issues such as the poor conductivity of sulfur and the discharge product Li_2_S and the severe dissolution of lithium polysulfides (LiPSs) seriously hinder the development of Li-S batteries. Among them, the “shuttle effect” caused by the dissolution of LiPSs is the most deadly, which directly leads to the reduction of battery capacity and Coulomb efficiency. How to suppress the “shuttle effect” has become the key direction to be overcome in the application of Li-S batteries [14,15,16,17].

In general, accelerating the conversion of LiPSs to Li_2_S and anchoring LiPSs to prevent their migration to the lithium metal anode can effectively alleviate the shuttle effect. Separator modification is considered to be a proven solution because it meets both of the above functional requirements [18,19]. Due to the advantages of good conductivity, light weight, and stable physicochemical properties, porous carbon skeleton materials, such as graphene, mesoporous/microporous carbon, carbon nanotubes, and carbon nanofibers, are often used as typical separator modification host materials, whose three-dimensional network structure can provide accommodation space for the volume expansion of the sulfur cathode during cycling [15,20,21]. In addition, for the shuttle effect, their suitable porosity can exert a better physical adsorption effect on LiPSs during the charging and discharging process. However, conventional carbon materials are non-polar molecular structures, and a single reliance on physical adsorption to capture polysulfides to mitigate the shuttle effect is unsatisfactory [22]. Therefore, the introduction of polar groups for selective adsorption of LiPSs by bonding to their S atoms has become the preferred choice for separator modification schemes [23,24]. For example, polar materials containing Mo and Ti can target and adsorb polysulfides and catalyze their conversion while maintaining their own structural integrity [25,26]. Among them, compounds of molybdenum are highly metallic, and by combining with highly conductive carbon materials, their highly conductive networks can provide ample transport channels for ions and substantially increase the number of active sites during electrochemical reactions [27,28,29,30]. For Li-S batteries, reports show that the strong polarity of molybdenum compounds can significantly attract and deflect polysulfide electron pairs, resulting in strong adsorption to them and effective cut-off of the shuttle effect by facilitating their discharge conversion. In addition, due to the acceleration of the key step (Li_2_S_2_~Li_2_S), the nucleation of Li_2_S will be effectively improved, and the reduction of Li_2_S stacking can enhance the reversibility of its decomposition during charging and improve the utilization of lithium [25,31].

As a typical molybdenum compound, molybdenum carbide (Mo_2_C), with an electronic structure similar to that of platinum, possesses ultra-low electrical conductivity (1.41 × 10^4^ S cm^−1^) and high electrochemical activity, which makes it exhibit good catalytic performance in electrochemical reactions such as in lithium–oxygen batteries and oxygen evolution/reduction reactions [32,33]. For example, carbon-encapsulated Mo_2_C nanoparticles and carbon nanotubes co-loaded with nickel foam as a binder-free cathode exhibited low overpotential and excellent cycling stability in lithium–oxygen batteries [34]. In addition, the combination of Mo_2_C with transition metals can effectively reduce the amount of unoccupied Mo and further enhance hydrogen evolution reaction performance [35]. It is not difficult to infer that the introduction of Mo_2_C will be expected to bring a stronger polarity to the carbon-based cathode material of Li-S batteries and play a better role in facilitating the catalytic conversion of LIPSs, thus alleviating the shuttle effect.

Here, we prepared Mo_2_C-loaded porous carbon nanosheet composites (Mo_2_C/C) by a facile aging synthetic process and applied them for Li-S battery separator modification. As a carbon-based composite with a polar catalyst, Mo_2_C/C can effectively adsorb and promote the reversible conversion of LiPSs, thereby suppressing the shuttle effect and improving the electrochemical performance of the battery. The Li-S battery with a Mo_2_C/C separator has high specific capacities of 1470 and 799 mAh g^−1^ at 0.1 and 2 C, respectively, showing a good rate performance. In addition, the capacity retention after 400 cycles at 1 C is up to 62%, with a decay of 0.09% per cycle, thus showing good cycle stability.

## 2. Experimental Section

### 2.1. Preparation of Mo_2_C/C

Exactly 7.15 g of ammonium molybdate tetrahydrate (NH_4_)_6_Mo_7_O_24_·4H_2_O and 6 g of agar powder were added to 100 mL of deionized water and dissolved at 80 °C with stirring, and a gel-like product was obtained after cooling. Subsequently, 100 mL of a solution dissolved with 9.6 g NH_4_HCO_3_ was poured on the surface of the gel and aged at room temperature for 48 h. Finally, the aged gel was placed in a tube furnace and heated at a rate of 10 °C per minute under nitrogen protection to 900 °C and then held for 6 h. A Mo_2_C/C sample was obtained after natural cooling.

### 2.2. Preparation of the Cathode

The sublimated sulfur and acetylene black were well ground and mixed in a mass ratio of 7:3, covered with aluminum foil, and held in a tube furnace at 155 °C for 12 h under an argon atmosphere to obtain a sulfur cathode. Subsequently, the cathode material, polyvinylidene fluoride (PVDF), and acetylene black (mass ratio: 7:2:1) were ground and mixed in a mortar and poured into a sealed glass vial, stirred with an appropriate amount of N-methyl-2-pyrrolidinone (NMP) to obtain a homogeneous slurry, which was uniformly coated onto aluminum foil with a 200 μm squeegee, vacuum-dried overnight, and finally cut into 15 mm disc electrodes. In addition, a high sulfur load of 3.84 mg/cm^2^ was prepared in the same way. The E/S ratio used for low loading in this study was 20 μL/mg. The E/S ratio used for high loading was 10 μL/mg.

### 2.3. Preparation of the Modified Separators

First, Mo_2_C/C, Ketjen black and PVDF were mixed well in NMP solvent in a mass ratio of 7:2:1 to prepare a homogeneous slurry. Subsequently, the slurry was cast onto a Celgard2500 separator with a 25 μm squeegee and dried at 80 °C. Finally, the coated separator was cut into discs of 19 mm and stored under vacuum before use.

### 2.4. LiPSs Adsorption

A 5 mM solution of Li_2_S_6_ was prepared as a mock solution of LiPSs by dissolving S_8_ and Li_2_S in DME at a molar ratio of 5:1 with stirring, and the dissolution process was kept at a constant temperature of 60 °C and carried out in an argon-filled glove box. Then, a 10 mg Mo_2_C/C sample or carbon was added to 4 mL of the as-obtained Li_2_S_6_ solution and left for 12 h. Finally, the supernatant was studied using a UV spectrophotometer to evaluate the adsorption capacity of Mo_2_C/C or carbon on LiPSs.

### 2.5. Symmetrical Cell Test

First, Mo_2_C/C or C, super P, and PVDF were mixed in a mass ratio of 8:1:1 using NMP as the solvent to prepare a homogeneous slurry, and then the slurry was cast on aluminum foil, vacuum-dried, and cut into 12 mm discs as electrodes. A 0.2 M Li_2_S_6_ electrolyte was synthesized by dissolving S and Li_2_S in DME at a mass ratio of 5:1 and then stirring at 60 °C overnight in an argon-filled glove box. The two obtained electrodes were then loaded into CR2032 coin-type cells with PP separators and 20 μL as-obtained Li_2_S_6_ was added as the electrolyte, respectively, and the CV of the symmetric cell was tested under a potential window of −1.0 to 1.0 V to the study redox reaction of LiPSs on Mo_2_C/C.

### 2.6. Li_2_S Precipitation Experiment

Li_2_S precipitation experiments were used to study the liquid–solid conversion kinetics. A 0.5 M Li_2_S_8_ electrolyte was prepared by dissolving S and Li_2_S in tetraethylene glycol dimethyl ether in a 7:1 mass ratio. The CR2032 coin cell assembled for the test used Celgard2500 as the separator, lithium foil as the anode, Mo_2_C/Carbon as the cathode, and 30 μL of the above Li_2_S_8_ solution as the electrolyte. The cell was first discharged to 2.06 V, and then Li_2_S nucleation and growth were performed using a 2.05 V potentiostatic discharge procedure until the current was less than 0.01 mA.

## 3. Results and Discussion

Figure 1 shows the preparation process of Mo_2_C/C. Deionized water is used as the solvent for the reaction, and ammonium molybdate and carbon source agar powder are dissolved by heating and stirring to form a homogeneous colloidal solution. Subsequently, pre-prepared amine bicarbonate solution was poured into the cooled colloidal solution, aged for 48 h, and then freeze-dried. Finally, the freeze-dried samples were annealed at a high temperature under argon protection to obtain porous carbon nanosheets loaded with Mo_2_C nanoparticles. Among them, the pre-placed ammonium bicarbonate will decompose at high temperature to release a sufficient amount of carbon dioxide to induce the formation of a large number of pore structures in the gel matrix, thus promoting the effective exposure of the catalyst active sites and allowing to accommodate the volume expansion of the cathode during the charging and discharging process.

Figure 2a shows the X-ray diffraction (XRD) pattern of Mo_2_C/C. The diffraction peaks at 34.3, 37.9, 39.4, and 61.5° correspond to the (100), (002), (101), and (110) crystal planes of Mo_2_C (JCPDS No. 35-0787) respectively, which indicates the successful preparation of Mo_2_C/C. Figure 2b,c shows the SEM images of the agarose gel precursor and the corresponding Mo_2_C/C material. The framework of the sample is built up from an interwoven web of carbon matrix. It can be seen that the carbon matrix obtained after annealing is constructed by carbon nanosheets with a size around 100 nm and possesses an obvious porous structure. A sufficient amount of pore structures can provide flexible channels for the mass transfer process during the electrochemical reaction and ensure sufficient exposure of the catalyst active site and effective adsorption of LiPSs, making the suppression of the shuttle effect possible. Figure 2d,e shows the high-resolution transmission electron microscopy (HRTEM) images of Mo_2_C/C. It can be seen from the figures that the size of the nanosheets is relatively uniform and in good agreement with the SEM observations. The lattice fringes of 0.236 nm shown in Figure 2e can be assigned to the (0 0 2) crystal plane of Mo_2_C which can be used as a typical representative to illustrate that the diameter of Mo_2_C is about 20 nm. Appendix A shows the element mapping of Mo_2_C/C, illustrating the uniform distribution of C, O, Mo, and N elements. Figure 2f shows the N_2_ absorption and desorption isotherms of Mo_2_C/C materials and their corresponding pore size distribution. The specific surface area of the Mo_2_C/C composite is determined to be 73 cm^2^ g^−1^ with an average pore size of 5 nm, confirming that the sample has a large number of mesopores and an appreciable specific surface area. Such a high surface area provides more surface active sites, which will certainly lead to improved electrochemical performance. The modification of the separator of Li-S batteries with Mo_2_C/C as a coating can effectively improve the wetting ability of the electrolyte on the separator, as shown in Figure 2g. For the Mo_2_C/C@PP-modified separator, the contact angle of the electrolyte is close to 0°, while the contact angle of the electrolyte on the original PP separator is 46°, indicating that the introduction of the Mo_2_C/C coating is expected to enhance the flux of lithium-ion and electron transport, thus reducing polarization and improving electrochemical performance. SEM images of the cross-section of the Mo_2_C/C@PP-modified separator are shown in Figure 2h. The thickness of the modified coating is about 25 μm, and it remains in close contact with the PP separator even after cropping. In addition, the surface coating of the modified separator shows no significant peeling and flaking in the separator bending test and the immersion test, further demonstrating the good mechanical stability of the modified separator (Figure 2i). The elemental valence state and chemical composition of Mo_2_C/C were further investigated by X-ray photoelectron spectroscopy (XPS). Appendix A shows the full XPS spectrum of the Mo_2_C/C composite. The peaks at 231.8, 284.08, 393.08, 419.8, and 528 eV correspond to Mo 3d, C1s, Mo 3p_3/2_, Mo 3p_1/2_, and O 1s [36], respectively, in addition to no obvious impurity peaks, thus confirming the existence of Mo, C, and O elements, which is consistent with the EDX energy spectrum. In the Mo 3d spectrum in Figure 2j, the fitted peaks at 228.5 and 231.7 eV correspond to Mo^2+^ in Mo_2_C, respectively, indicating the successful preparation of Mo_2_C [37]. The fitted peaks at 236 and 232.7 eV coincide with the Mo^6+^ 3d_3/2_ and 3d_5/2_ orbitals in MoO_3_ [38], respectively, and the fitted peaks at 233.4 and 299.8 eV can be assigned to the Mo^4+^ 3d_3/2_ and 3d_5/2_ orbitals in MoO_2_, respectively. The existence of MoO_2_ and MoO_3_ may originate from the surface oxidation of Mo_2_C. The C 1s from Figure 2k can be resolved into three peaks: the fitted peak at 283.5 eV indicates the Mo–C bond in Mo_2_C and the remaining fitted peaks located at 284.8 and 286.02 eV can be attributed to the C–C bond and the C–O–Mo bond, respectively. For the O 1s spectrum in Figure 2l, the fitted peaks at 530.7 and 532.56 eV correspond to the C–O and Mo–O bonds, respectively, which may originate from the surface oxidation of the carbon matrix and Mo_2_C [39]. The XPS analysis and XRD results echo each other and both point to the successful synthesis of Mo_2_C/C composites after calcination.

To verify the adsorption capacity of Mo_2_C/C on LiPSs, we added equal amounts of Mo_2_C/C and its blank control carbon nanosheets to equal volume of 5 mM Li_2_S_6_ solution for the adsorption experiments. As shown in Figure 3a, after 12 h of static adsorption, the supernatant corresponding to Mo_2_C/C adsorption appeared colorless and transparent, while the solution treated with carbon nanosheets left a distinct brown color. The UV spectrum shows that the peak intensity at 265 nm of the supernatant is greatly reduced after Mo_2_C/C adsorption, indicating that the content of Li_2_S_6_ in the solution is greatly reduced, which fully verifies the strong adsorption effect of Mo_2_C/C on Li_2_S_6_ [22]. Adsorption is a prerequisite for catalysis, and for the specific catalytic ability of Mo_2_C/C to promote the reversible conversion of LiPSs, we evaluated it by assembling symmetric cells with Mo_2_C/C materials as electrodes and Li_2_S_6_ solutions as electrolytes and testing their CV characterization. The CV curves of the symmetric cell at different scan rates from 1 to 5 mv s^−1^ are shown in Figure 3b. The reduction peaks located at −0.453 V (peak a) and −0.74 V (peak b) can be attributed to the continuous reduction of Li_2_S_6_ to Li_2_S_2_/Li_2_S, while the peaks a’ and b’ at 0.465 V and 0.737, respectively, correspond to the continuous conversion of Li_2_S_2_/Li_2_S to S_8_. In addition, it can be observed that there is a small shift of the redox peak with the increase in the scan rate, which may be due to the polarization phenomenon caused by the increase in the scan rate and the consumption of Li_2_S_6_. However, the redox peaks remain with the increase in the scan rate, which indicates the presence of rapid redox reactions at the Mo_2_C/C interface [40]. Furthermore, we tested the CV curves of the Mo_2_C/C symmetric cell with a conventional Li-S battery electrolyte as a substitute and failed to observe a significant peak current (Figure 3c), which confirms that Mo_2_C/C can indeed promote the redox reaction of LiPSs. Figure 3d shows the multi-loop CV curves of the Mo_2_C/C symmetric cell. After the first activation cycle, the subsequent largely overlapping curves illustrate the high reversibility of the catalytic conversion of LiPSs by Mo_2_C/C [41].

Based on the above characterization, modification of the cathode-side separator of the Li-S battery with a Mo_2_C/C coating is expected to improve the contact interface between the cathode and the separator, thereby accelerating lithium ions transport. To verify the specific promotion effect, we assembled cells with different separators and calculated the lithium ions diffusion coefficients based on the CV curves for different scan rates. Figure 4a shows the CV curves of the battery with a Mo_2_C/C-modified separator at scan rates of 0.1–0.5 mV s^−1^. Obviously, the reduction peaks are gradually shifted toward the negative voltage and the oxidation peak is gradually shifted toward the positive voltage as the scan rate increases, which may be due to the polarization associated with ion transfer.

As shown in Figure 4b–d, all of the peak currents are linearly related to the square root of the scan rate, and the diffusion coefficient of lithium ions can be obtained from the Randles–Sevcik equation as follows:*I_P_* = (2.69 × 10^5^) *n*^1.5^*S D*^0.5^
*C V*^0.5^(1)
where Ip, D, and V are the peak current, lithium ions diffusion coefficient, and scan rate, respectively, while N, S, and C are constants representing the number of electrons involved in the electrochemical reaction, the effective electrode area, and the lithium concentration in the electrolyte, respectively. It can be deduced from the equation that the slope of Ip/V0.5 is positively correlated with D. Obviously, the absolute value of D is the largest for the Mo_2_C/C-modified separator compared to the PP and carbon-based modified separators, which is most likely due to the increased wettability of the electrolyte to the separator, resulting in effective enhancement of the diffusion process of lithium ions within the battery.

To further investigate the effect of the catalytic properties of Mo_2_C on the liquid–solid conversion process of LiPSs, a potentiostatic discharge experiment was carried out. Prior to the nucleation process, long-chain LiPSs were converted to short-chain LiPSs by discharging to 2.06 V, followed by an extremely small current to drive the nucleation and growth of lithium sulfides (Li_2_S) [42]. Here, the nucleation ability of Li_2_S can be assessed by calculating the integral area of the loop curve under the current curve. The comparison shows that the battery possessing the Mo_2_C/C-modified separator has a shorter Li_2_S nucleation time and stronger Li_2_S deposition ability, as seen in Figure 4e,f.

From the above results, it can be inferred that the Mo_2_C/C separator contributes to the enhancement of interfacial compatibility at the cathode side due to its good wettability and high electrolyte absorption, which will shorten the electrolyte filling time, promote lithium-ion migration, induce uniform deposition of solid Li_2_S_2_/Li_2_S, and reduce passivation at the electrode/electrolyte active interface. In addition, sufficient polar electrocatalytic active sites on its surface can effectively promote the adsorption and catalytic conversion of soluble lithium polysulfide, provide additional electron pathways for the sulfur cathode, suppress the shuttle effect, realize the recycling of sulfur, and improve energy efficiency.

The effect of Mo_2_C/C as a separator modification material on the specific electrochemical properties of Li-S batteries was investigated by assembling 2032-type coin cells. Figure 5a displays the initial CV curves of cells with different modified separators at 0.1 mV s^−1^. For the cell with the Mo_2_C/C@PP separator, the reduction peaks at 2.0 and 2.26 V correspond to the transformation of sulfur to soluble LiPSs (Li_2_S*_x_*, 4 ≤ *x* ≤ 8) and soluble polysulfides to solid Li_2_S_2_/Li_2_S, respectively, while the oxidation peak at 2.4 V points to the oxidation of short-chain Li_2_S/Li_2_S_2_ to long-chain Li_2_S_x_ and finally to S_8_. In addition, the cell with the Mo_2_C/C@PP separator exhibited the largest integrated area of the CV curve, indicating stronger kinetics of the redox reaction and more favorable interconversion of LiPSs and Li_2_S during charging and discharging, which may be attributed to the good adsorption and catalytic conversion of LiPSs by Mo_2_C/C. Figure 5b,c shows the detailed charge–discharge curves of the three separators at 0.1 C. For the Mo_2_C/C material, the release capacity is 400.7 mAh g^−1^ in the stage from S_8_ to Li_2_S_4_, which is higher than that of the carbon matrix (329.5 mAh g^−1^) and PP (397.9 mAh g^−1^), and the first plateau of the battery is around 2.3 V, which implies higher sulfur utilization and fast liquid–liquid conversion. In addition, the interfacial energy barrier of the Mo_2_C/C battery is 32.8 mV, which is smaller than that of the carbon matrix (36.9 mV) and PP (45.2 mV) batteries, as well as a lower onset potential. This further confirms that Mo_2_C with high conductivity and high catalytic properties can reduce battery polarization and effectively promote the conversion of long-chain LiPSs to short-chain LiPSs, promotes the nucleation and growth of Li_2_S, and improves the utilization of the sulfur cathode [43,44].

The cycling performance of the batteries with the three separators at 0.2 C is shown in Figure 5d. The battery with the Mo_2_C/C separator presents specific capacities of 1151 and 913 mAh g^−1^ for the 1st and 100th cycles, respectively, with a stable Coulombic efficiency close to 99%, thus showing good cycling stability. For the cells with the carbon-based and pristine PP separators, the initial specific capacities are 982 and 906 mAh g^−1^, respectively, and their specific capacities maintain at 745 and 712 mAh g^−1^ after 100 cycles. Figure 5e shows the rate capabilities of the three separators at different current densities. For the Mo_2_C/C separator, it exhibits specific capacities of 1470, 1056, 967, 883, and 799 mAh g^−1^ at 0.1, 0.2, 0.5, 1, and 2 C, respectively. Notably, when the current density is switched back to 0.2 C, the battery still remains at a discharge capacity of 1041 mAh g^−1^, thus revealing excellent rate performance and high-capacity reversibility. In comparison, the capacities of the carbon-based modified separator and the pristine PP separator are 597 and 224 mAh g^−1^ at 2 C, respectively. Figure 5f shows the charge–discharge curves of the Mo_2_C/C-modified separator corresponding to different rates. It can be seen that the charge–discharge curves maintain an obvious plateau even when the current gradually increases, which indicates that Mo_2_C nanoparticles can effectively ensure the full utilization of sulfur to obtain a higher specific capacity. The recycling capacity of Li-S batteries under high sulfur load is an important indicator to judge whether the modified materials can be commercialized. In general, the higher the sulfur loading, the more difficult it is to ensure the utilization of active substances in an equivalent amount of electrolyte [45]. For verification, we assembled a Li-S battery with a cathode containing 3.84 mg cm^−2^ of sulfur and tested its cycling performance with an electrolyte/activator of 10 μL/mg, as shown in Figure 5g. The initial discharge capacity of the battery was 833 mAh g^−1^ at 0.2 C, and the capacity was 772 mAh g^−1^ after 120 cycles, with a capacity retention rate of 92%. This indicates that the Mo_2_C/C material has good electrochemical performance even under the harsh conditions of high sulfur load and a low amount of electrolyte, thus demonstrating its potential for practical production applications. The Li-S batteries applying Mo_2_C/C separators exhibited higher specific capacities than the other modified separators, such as MWCNTs/NCQDs (895 mAh g^−1^), MnO_2_/CNT (794.2 mAh g^−1^), and KB/Mo_2_C (664 mAh g^−1^) [40,46,47], especially in terms of their capacity decay rate, as summarized in Appendix A.

The long-cycle performance of Li-S batteries is one of the crucial performance indicators. Figure 5h shows 400 cycles of the battery with the Mo_2_C/C@PP separator at 1 C. The battery exhibits high specific capacities of 930 and 585 mAh g^−1^ for the 1st and 400th cycles at 1 C, respectively, with a capacity decay of only 0.09% per cycle and maintains a Coulombic efficiency of 97%. Appendix A visualize the morphological changes of different separators before and after 100 cycles. For the Mo_2_C/C separator, some precipitated adsorbates exist on the surface of the separator after several cycles, but it still maintains good porosity, indicating that it could better adsorb and catalyze the reversible conversion of LiPSs during the charging and discharging process. In contrast, the carbon separator does not show significant changes before and after cycling, indicating that it may have no special role in promoting the electrochemical reaction process of Li-S batteries. Appendix A shows the Nyquist plots of Mo_2_C/C-modified, carbon-based, and pristine PP separators with similar contours, consisting of high-frequency straight lines and low-frequency semicircles, corresponding to charge transfer impedance and ions diffusion resistance. The results show that the separator modified with Mo_2_C/C has a smaller diameter semicircle, which indicates less charge transfer impedance.

## 4. Conclusions

In conclusion, we prepared Mo_2_C nanoparticle-supported carbon nanosheet material (Mo_2_C/C) by the sol–gel method and used it as an efficient electrocatalyst for the modification of a Li-S battery separator. The porous structure of carbon nanosheets can effectively provide buffer space for the volume expansion of the sulfur cathode during cycling and act as a physical barrier to inhibit the shuttle effect. As a polar material, Mo_2_C can effectively adsorb LiPSs and accelerate their reversible conversion during the charge–discharge process. Meanwhile, its high electrical conductivity facilitates the acceleration of lithium-ion and electron transport. As a result, a Li-S battery with a Mo_2_C/C-modified separator has high specific capacities of 1470 and 799 mAh g^−1^ at 0.1 and 2 C, respectively, showing a good rate capability. After 400 cycles at 1 C, the capacity retention rate is 62% with a decay of 0.09% per cycle, thus showing good cycling stability. Moreover, even with sulfur loading of 3.84 mg cm^−2^, capacity retention of 92% is achieved after 120 cycles at 0.2 C, thus revealing their good potential for practical applications.

## Figures and Tables

**Figure 1 materials-16-01635-f001:**
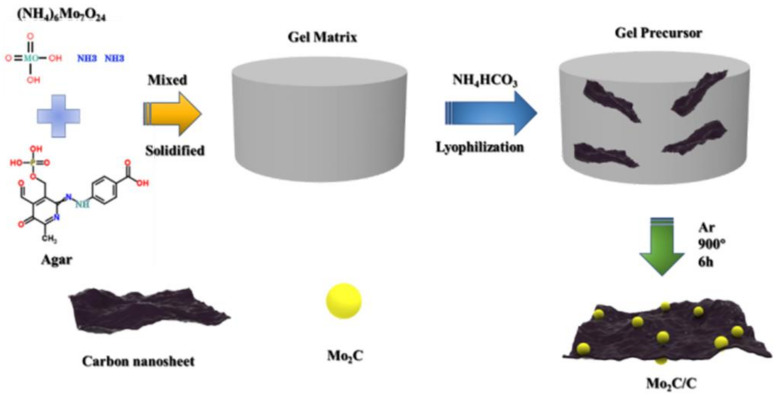
Schematic diagram of the preparation process of Mo_2_C/C.

**Figure 2 materials-16-01635-f002:**
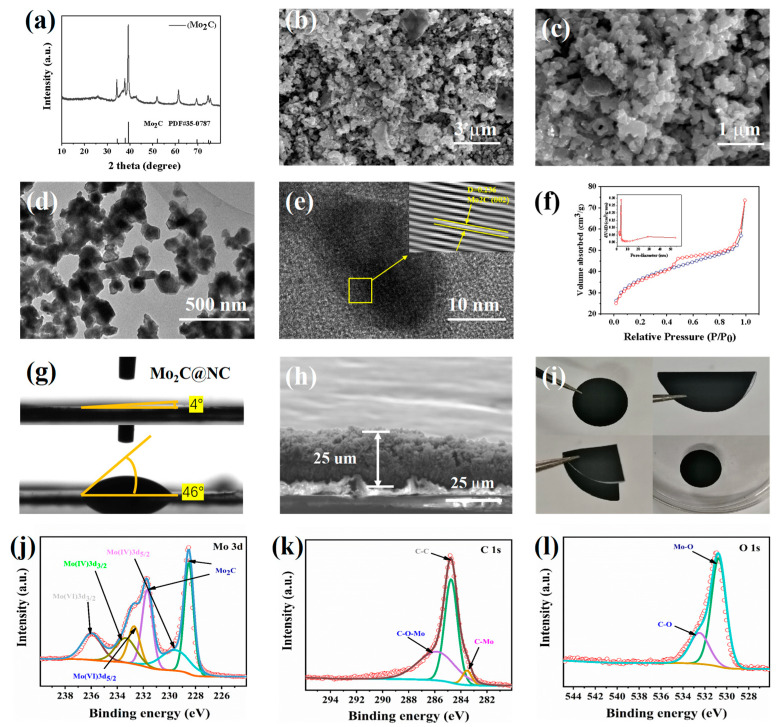
(**a**) XRD pattern of Mo_2_C/C. (**b**,**c**) SEM images of the agarose gel precursor and the corresponding Mo_2_C/C. (**d**,**e**) HRTEM images Mo_2_C/C. (**f**) BET and pore size distribution of Mo_2_C/C. (**g**) Electrolyte contact angle on the original PP and Mo_2_C/C@PP separators. (**h**) Cross-sectional view of the Mo_2_C/C@PP separator. (**i**) Digital photos of Mo_2_C/C separators under different test conditions. (**j**) Full spectrum, (**k**) Mo 3d, (**l**) C 1s, and (**d**) O 1s.

**Figure 3 materials-16-01635-f003:**
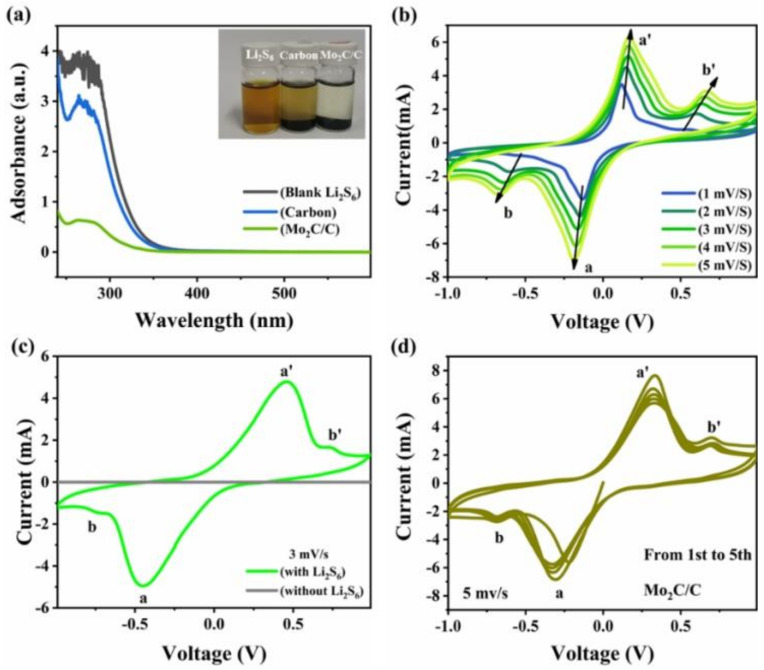
(**a**) UV–vis spectra of the supernatant after adsorption of Li_2_S_6_ solutions by different adsorbents. Inset: Digital photo of the corresponding Li_2_S_6_ solutions. (**b**) CV curves of the symmetric battery with the Mo_2_C/C as electrodes and Li_2_S_6_ solution as an electrolyte at various scan rates. (**c**) CV comparison of symmetric cells assembled with the conventional electrolyte and Li_2_S_6_ solution and Mo_2_C/C electrodes at 3 mV s^−1^. (**d**) CV cycling curves of the symmetric cell with Mo_2_C/C electrodes at 5 mV s^−1^.

**Figure 4 materials-16-01635-f004:**
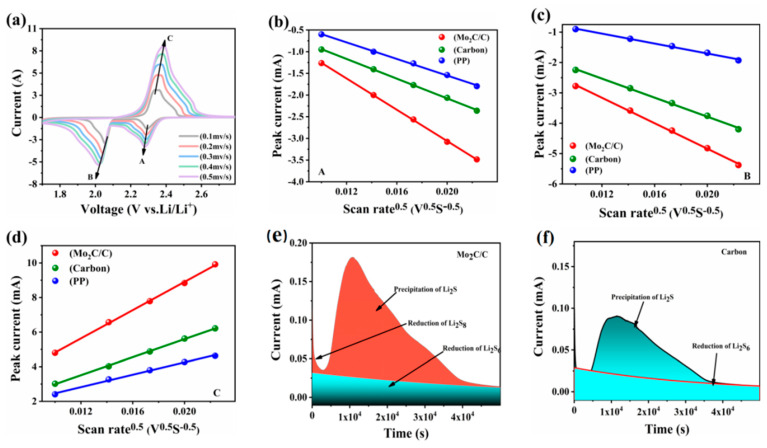
(**a**) CV curves of the battery with a Mo_2_C/C-modified separator at scan rates ranging from 0.1 to 0.5 mV s^−1^. (**b**–**d**) Relationship between CV peak currents and scan rates for LSBs using different separators. (**e**,**f**) Chronoamperometry results of constant current and constant voltage discharge at 2.05 V on Mo_2_C/C and a carbon nanosheet.

**Figure 5 materials-16-01635-f005:**
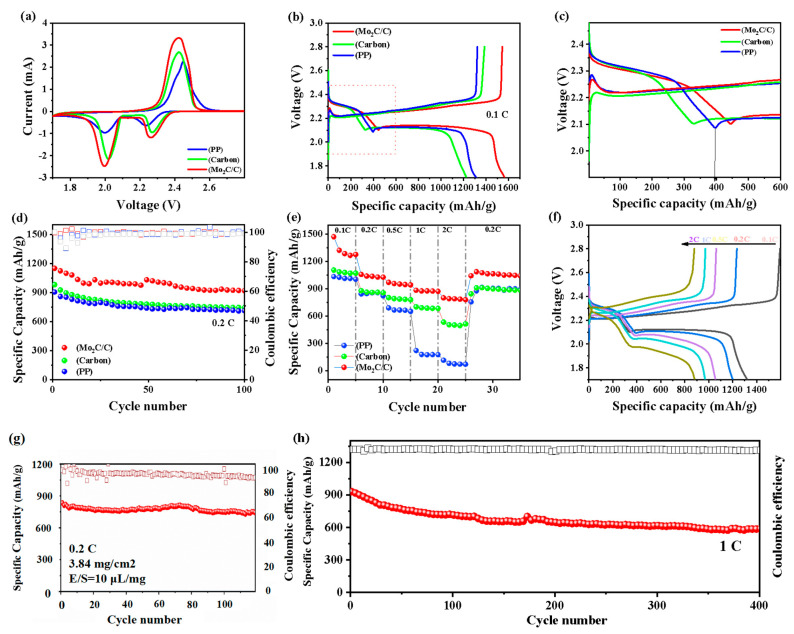
(**a**) First CV curves of different separators at 0.1 mV s^−1^. (**b**) Galvanostatic charge–discharge curves of different separators at 0.1 C. (**c**) The details of the charge–discharge curves at 0.1 C. (**d**) Cycling performance of different separators at 0.2 C. (**e**) Rate capabilities of different separators. (**f**) Charge–discharge curves corresponding to different rates. (**g**) Cycling performance of the Mo_2_C/C separator under sulfur loading of 3.84 mg cm^−2^. (**h**) Long cycle performance of the Mo_2_C/C separator at 1 C.

## Data Availability

The data used to support the findings of this study are available from the corresponding author upon request.

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
