# Peer review of "Mo2C-Loaded Porous Carbon Nanosheets as a Multifunctional Separator Coating for High-Performance Lithium–Sulfur Batteries"

_materials, 2023, doi:10.3390/ma16041635_

Round 1

Reviewer 1 Report

In this manuscript, authors prepared Mo2C-loaded porous carbon nanosheet composites (Mo2C/C) by a facile aging synthetic process and applied them for Li-S battery separator modification. The Li-S battery with Mo2C/C separator has high specific capacities and a good rate performance. The experiments are well-conducted and interesting results obtained. In my opinion, I accept the manuscript in present form.

Author Response

Thank you for reviewing our manuscript and acknowledging our work, and thank you again for accepting our current manuscript.

Reviewer 2 Report

Summary:

This Research Paper materials-2172673 titled, “Mo2C-loaded porous carbon nanosheets as a multifunctional separator coating for high-performance lithium-sulfur batteries,” reports the extended study of the modified separator for the use in the electrochemical stabilization of lithium-sulfur batteries. The separator modification uses Mo2C nanoparticles-embedded carbon nanosheet matrix material, which is coated on a commercial separator. The coating layer reduces the loss of polysulfides by blocking their diffusion. The trapped material is catalyzed by the Mo2C for showing a rate performance with high specific capacities of 1470 and 799 mAh g-1 at C/10 and 2C rates, respectively.

General comment:

This paper reports the material synthesis and application in the lithium-sulfur battery separator. However, some necessary revisions and corrections are requested. Minor revisions are suggested to provide the critical data and parameters. Hope the authors feel the comment useful.

Comments:

(1) In all, the manuscript and analysis are suggested to be improved and polished. In the introduction and the comparison analysis, Mo2C/C is the material highlighted. It is necessary to have Mo2C as a reference to compare with carbon and Mo2C/C for a reasonable discussion. Figure 5 needs to arrange the plots that are currently at the wrong places. Moreover, the long cycle performance of the Mo2C/C separator at 1 C (Figure 1h) is in an incorrect Y-/X-axis ratio. The cycling performance of the Mo2C/C separator under sulfur loading of 3.84 mg cm-2 is a good data that is more important than all others. However, Figure 5j is not shown. In addition, it is curious to know that the performance “the carbon matrix (371.1 mAh g-1) and PP (362.4 mAh g-1)” is perfectly the same as a published paper “Ni-Co@C//PP (371.1 mAh g−1) and PP (362.4 mAh g−1)” till the decimal point. Why? (Reference: Implanting nickel and cobalt phosphide into well-defined carbon nanocages: A synergistic adsorption-electrocatalysis separator mediator for durable high-power Li-S batteries, 10.1016/j.ensm.2021.03.026)

[Suggestion] Please revise and check the manuscript and data.

(2) In the Experimental section, the thickness and mass loading of the coating materials are necessary to be reported. The sulfur loading and content of the cell with the consideration of the weight of coating material are necessary to be reported.

[Suggestion] Please report the suggested information if it is possible to collect from the original papers.

(3) In the material analysis, the cross sectional SEM image shows the thickness of the coating layer is incorrect. The scale bar and the reference thickness bar tell different information. Moreover, it is also suggested to include the separator as a reference for 25um.

[Suggestion] Please check and correct the thickness of the modified separator.

(4) In electrochemical analysis, the CV curves of the battery with Mo2C/C modified separator show a pair of cathode and anodic peaks at the end the discharge and charge. The discharge peaks at 1.9 V and charge peaks at 2.5 V are suggested to be identified and analyzed. Moreover, it is curious that the CV curves of the battery with Mo2C/C modified separator show different data in Figure 4a and 5a. In addition, the discharge/charge efficiency is necessary data in the rate performance.

[Suggestion] Please revise the manuscript by considering the suggested writing modification.

(5) In all, the engineering design and function of the modified separator in the lithium-sulfur cells are necessary for the discussion and the introduction. To support the discussion, some pioneer and recent works are suggested to show the research trends and progresses: (ACS Nano, 2015, 9, 3002-3011: initial works in modified lithium-sulfur battery separator; Sustainable Energy Fuels, 2021, 5, 5656-5671; Membranes 2022, 12, 790; Molecules 2022, 27, 228: summary of coating materials and functions of the modified separators)

[Suggestion] Please discuss the engineering design and material development of the modified separators in the introduction with the support of the suggested references.

Author Response

Point 1: Figure 5 needs to arrange the plots that are currently at the wrong places. Moreover, the long cycle performance of the Mo2C/C separator at 1 C (Figure 1h) is in an incorrect Y-/X-axis ratio. The cycling performance of the Mo2C/C separator under sulfur loading of 3.84 mg cm-2 is a good data that is more important than all others. However, Figure 5j is not shown. In addition, it is curious to know that the performance “the carbon matrix (371.1 mAh g-1) and PP (362.4 mAh g-1)” is perfectly the same as a published paper “Ni-Co@C//PP (371.1 mAh g1) and PP (362.4 mAh g1)” till the decimal point. Why? (Reference: Implanting nickel and cobalt phosphide into well-defined carbon nanocages: A synergistic adsorption-electrocatalysis separator mediator for durable high-power Li-S batteries, 10.1016/j.ensm.2021.03.026)

Response 1: Thank you very much for your suggestion. We have corrected the error in Figure 5 based on your comments. In addition, the more meaningful high-sulfur loading cycle performance has also been added to Figure 5. The specific modifications are as follows.

Figure 5 (a) First CV curves of different separators at 0.1 mV s-1. (b) Galvanostatic charge-discharge curves of different separators at 0.1 C. (c) The details of charge-discharge curves at 0.1 C. (d) Cycling performance of different separators at 0.2 C. (e) Rate capabilities of different separators. (f) Charge-discharge curves corresponding to different rates. (g) Cycling performance of Mo2C/C separator under sulfur loading of 3.84 mg cm-2. (h) Long cycle performance of the Mo2C/C separator at 1 C.

We apologize for the fatal error you raised regarding the specific capacities when applying the carbon matrix and PP separators, for which we did refer to the paper you mentioned and made the relevant error. We have carefully corrected the error as follows.

For the Mo2C/C material, the release capacity is 400.7 mAh g-1 in the stage from S8 to Li2S4, which is higher than that of the carbon matrix (329.5 mAh g-1) and PP (397.9 mAh g-1), and the first plateau of the battery is around 2.3 V, which implies a higher sulfur utilization and a fast liquid-liquid conversion. (Page 17, Line 4)

Point 2: In the Experimental section, the thickness and mass loading of the coating materials are necessary to be reported. The sulfur loading and content of the cell with the consideration of the weight of coating material are necessary to be reported.

Response 2: Thank you for your kind reminding. we have added it in our revised manuscript.

The sublimated sulfur and acetylene black were well ground and mixed in a mass ratio of 7:3, covered with aluminum foil and held in a tube furnace at 155 °C for 12 h under an argon atmosphere to obtain a sulfur cathode. Subsequently, the cathode material, polyvinylidene fluoride (PVDF) and acetylene black (mass ratio: 7:2:1) were ground and mixed in a mortar and poured into a sealed glass vial, stirred with an appropriate amount of N-methyl-2-pyrrolidinone (NMP) to obtain a homogeneous slurry, which was uniformly coated onto aluminum foil with a 200 um squeegee, vacuum dried overnight, and finally cut into 15 mm disc electrodes. In addition, a high sulfur load of 8 mg/cm2 was prepared in the same way. The E/S ratio used for low loading in this study was 20 μL/mg. The E/S ratio used for high loading was 10 μL/mg. (Page 5, Line 16)

Ponit 3: In the material analysis, the cross sectional SEM image shows the thickness of the coating layer is incorrect. The scale bar and the reference thickness bar tell different information. Moreover, it is also suggested to include the separator as a reference for 25 um.

Response 3: Thank you for your careful review, we have corrected the incorrect scale in Figure 2h, as detailed below.

Figure 2 (h) Cross-sectional view of the Mo2C/C@PP separato

Point 4: In electrochemical analysis, the CV curves of the battery with Mo2C/C modified separator show a pair of cathode and anodic peaks at the end the discharge and charge. The discharge peaks at 1.9 V and charge peaks at 2.5 V are suggested to be identified and analyzed. Moreover, it is curious that the CV curves of the battery with Mo2C/C modified separator show different data in Figure 4a and 5a. In addition, the discharge/charge efficiency is necessary data in the rate performance.

Response 4: Thank you for your kind reminding. We have added the identification and analysis of the discharge and charge peaks to our revised manuscript.

For the cell with Mo2C/C@PP separator, the reduction peaks at 2.0 and 2.26 V correspond to the transformation of sulfur to soluble LiPSs (Li2Sx, 4 ≤ x ≤ 8) and soluble polysulfides to solid Li2S2/Li2S, while the oxidation peak at 2.4 V points to the oxidation of short-chain Li2S/Li2S2 to long-chain Li2Sx and finally to S8, respectively. (Page 16, Line 8)

Indeed, there is some discrepancy between the CV data in Figure 4a and Figure 5a, which may be due to systematic errors in the electrochemical performance of different batches of assembled cells, but such systematic errors did not affect the relevant rate performance and the regularity of the comparison of the electrochemical performance under different separators, and we consider the error to be within a reasonable range.

Point 5: In all, the engineering design and function of the modified separator in the lithium-sulfur cells are necessary for the discussion and the introduction. To support the discussion, some pioneer and recent works are suggested to show the research trends and progresses: (ACS Nano, 2015, 9, 3002-3011: initial works in modified lithium-sulfur battery separator; Sustainable Energy Fuels, 2021, 5, 5656-5671; Membranes 2022, 12, 790; Molecules 2022, 27, 228: summary of coating materials and functions of the modified separators)

Response 5: Thank you for your kind reminding. We found the work mentioned by the reviewers to be extremely informative, and we have included relevant literature in the introduction to support our ideas.

  1. Chu, S.; Manthiram, A. Bifunctional separator with a light-weight carbon-coating for dynamically and statically stable lithium-sulfur batteries. Adv. Funct. Mater. 2014, 24, 5299-5306.
  2. Huang, J.; Zhuang, T.; Zhang, Q.; Peng, H.; Chen, C.; Wei, F. Permselective graphene oxide membrane for highly stable and anti-self-discharge lithium-sulfur batteries. ACS Nano 2015, 9, 3002-3011.
  3. Huang, J.; Sun, Y.; Wang, Y.; Zhang, Q. Review on advanced functional separators for lithium-sulfur batteries. Acta Chim. Sinica 2017, 75, 173-188.
  4. Wu, H.; Chen, L.; Chen, Y. A mini-review of advanced separator engineering in lithium metal batteries. Sustain. Energ. Fuels 2021, 5, 5656-5671.
  5. Zhang, B.; Sun, B.; Fu, P.; Liu, F.; Zhu, C.; Xu, B. A review of the application of modified separators in inhibiting the “shuttle effect” of lithium–sulfur batteries. Membranes 2022, 12, 790.
  6. Huang, Y.; Yen, Y.; Tseng, Y.; Chung, S. Module-designed carbon-coated separators for high-loading, high-sulfur-utilization cathodes in lithium–sulfur batteries. Molecules 2022, 27, 228.

Reviewer 3 Report

Review Report on Mo2C-loaded porous carbon nanosheets as a multifunctional separator coating for high-performance lithium-sulfur batteries

1.     In the electrochemical performance authors mentioned that This further confirms that Mo2C with high conductivity and high catalytic properties can reduce the battery polarization and effectively promote the conversion of long-chain LiPSs to short-chain LiPSs, and promotes the nucleation and growth of Li2S and improves the utilization of sulfur cathode. Is there any specific region for such conversion? What is the source and how does it contribute to the electrochemical performance?

2.     The detailed analysis for interfacial reactions is required.

3.     In figure 2b,c. As shown in the surface morphology . The sufficient amount of pore structure can provide flexible channels for the mass transfer process during the electrochemical reaction and ensure sufficient exposure of the catalyst active site and effective adsorption of LiPSs, making the suppression of shuttle effect possible. Authors should mention the approx. pore density?

4.     Authors should also compare their findings with other reported literature.

After addressing these points, the manuscript can be accepted for publication

Author Response

Point 1: In the electrochemical performance authors mentioned that This further confirms that Mo2C with high conductivity and high catalytic properties can reduce the battery polarization and effectively promote the conversion of long-chain LiPSs to short-chain LiPSs, and promotes the nucleation and growth of Li2S and improves the utilization of sulfur cathode. Is there any specific region for such conversion? What is the source and how does it contribute to the electrochemical performance?

Response 1: Many thanks for your kind reminding. For the catalytic performance of Mo2C, no specific catalytic region was observed from our experiments. However, Mo2C, as a highly conductive catalyst, does contribute to the catalytic conversion of Sulfur in lithium-sulfur batteries, and this conclusion can be obtained from the comparison of the electrochemical performance exhibited by lithium-sulfur batteries with different separator applications. We believe that the Mo2C/C modified separator can promote the electrochemical performance of lithium-sulfur batteries as follows:

During the discharge process, the monomeric S8 at the anode is first reduced to the long-chain polysulfide Li2S8, which is soluble in the electrolyte, and continuously reduced to produce the short-chain polysulfide Li2S2 and Li2S, which are insoluble in the electrolyte. The Mo2C/C modified separator can effectively reduce the nucleation overpotential of Li2S and promote the conversion of Li2Sx (4 ≤ x ≤ 8) to Li2S, indicating the strong polarity of Mo2C has stronger adsorption and catalytic ability for lithium polysulfide, which helps to improve the reaction kinetics of lithium polysulfide conversion and increase the lithium ion/electron transfer rate, ultimately realizing high capacity, high multiplicity and long-term cycling stability of Li-S batteries.

Point 2: The detailed analysis for interfacial reactions is required.

Response 2: Many thanks for your kind advice. In fact, for the interfacial reaction, the calculation of the diffusion coefficient as well as the potentiostatic discharge experiment were performed to confirm that the application of the modified separator enhances the efficiency of the interfacial reaction. Based on your suggestion, we have added a summary statement of the interfacial reaction.

From the above results, it can be inferred that the Mo2C/C separator contributes to the enhancement of interfacial compatibility at the cathode side due to its good wettability and high electrolyte absorption, which will shorten the electrolyte filling time, promote lithium ion migration, induce uniform deposition of solid Li2S2/Li2S, and reduce passivation at the electrode/electrolyte active interface. Ultimately, it can improve the kinetics of redox reaction at the interface and promotes the conversion of complex LiPS during charging/discharging. (Page 15, Line 14)

Point 3: In figure 2b,c. As shown in the surface morphology. The sufficient amount of pore structure can provide flexible channels for the mass transfer process during the electrochemical reaction and ensure sufficient exposure of the catalyst active site and effective adsorption of LiPSs, making the suppression of shuttle effect possible. Authors should mention the approx. pore density?

Response 3: Thank you for your comments. The pore structure of the material in this work is constructed by the inherent micropores of the nanosheets and the stacking between the nanosheets. We characterized the inherent micropores of the material and the results showed that their average pore size is 5 nm. However, the pore density formed by the stacking between the nanosheets could not be calibrated due to the limited observation means.

Point 4: Authors should also compare their findings with other reported literature.

Response 4: Thank you for your constructive comment. We have added a comparison of this work with other reported literature.

Li-S batteries applying Mo2C/C separators exhibited higher specific capacities than other modified separators, such as MWCNTs/NCQDs (895 mAh g-1), MnO2/CNT (794.2 mAh g-1) and KB/Mo2C (664 mAh g-1) [40, 46, 47], especially in terms of capacity decay rate, as summarized in Table S1. (Page 18, Line 17)

Modified separator

S loading

(mg cm-2)

Coating thickness (μm)

Coating loading

(mg cm-2)

Rate capacity

(mAh g-1)

Capacity decay rate/cycle number/

C rate

Ref.

N,S-Mo2C/C-ACF //PP

1

6

0.5

900 (1C)

0.08%/600/1C

[1]

KB/Mo2C //PP

1.2

25

0.6

664 (1C)

0.076%/ 600/1C

[2]

Ni-Co-P@C//PP

1.8

3.2

0.4

906 (1C)

0.056%/1000/0.5C

[3]

FeP//PP

1.9-2.0

15.6

1.3

791.1 (1C)

0.038%/1500/1C

[4]

MnO2/CNT//PP

0.8

10

0.35

794.2 (1C)

0.136%/500/1C

[5]

VN//PP

1.6

25

1.52

760 (2C)

0.077%/800/1C

[6]

CuS/G//PP

1.85

19.5

0.5

568 (3C)

0.19%/200/1C

[7]

MWCNT s/NCQDs//PP

1.3-1.5

-

0.015

895 (1C)

0.056%/1000/1C

[8]

Mo2C/C//PP

1-1.4

20

0.5

930 (1C)

0.09%/400/1C

This work

Table S1. Electrochemical performances of this work compared with previous works involving different separators in recently reported literature.
